# Correlation with Apoptosis Process through RNA-Seq Data Analysis of Hep3B Hepatocellular Carcinoma Cells Treated with *Glehnia littoralis* Extract (GLE)

**DOI:** 10.3390/ijms25179462

**Published:** 2024-08-30

**Authors:** Min-Yeong Park, Sujin Lee, Hun-Hwan Kim, Se-Hyo Jeong, Abuyaseer Abusaliya, Pritam Bhangwan Bhosale, Je-Kyung Seong, Kwang-Il Park, Jeong-Doo Heo, Meejung Ahn, Hyun-Wook Kim, Gon-Sup Kim

**Affiliations:** 1Research Institute of Life Science and College of Veterinary Medicine, Gyeongsang National University, Gazwa, Jinju 52828, Republic of Korea; lilie17@daum.net (M.-Y.P.); shark159753@naver.com (H.-H.K.); tpgy123@gmail.com (S.-H.J.); yaseerbiotech21@gmail.com (A.A.); shelake.pritam@gmail.com (P.B.B.); kipark@gnu.ac.kr (K.-I.P.); 2Research Institute of Molecular Alchemy, Gyeongsang National University, 501, Jinju-daero, Jinju 52828, Republic of Korea; kssj9819@gnu.ac.kr; 3Laboratory of Developmental Biology and Genomics, BK21 PLUS Program for Creative Veterinary Science Research, Research Institute for Veterinary Science, College of Veterinary Medicine, Seoul National University, Seoul 08826, Republic of Korea; snumouse@snu.ac.kr; 4Biological Resources Research Group, Gyeongnam Department of Environment Toxicology and Chemistry, Korea Institute of Toxicology, 17 Jegok-gil, Jinju 52834, Republic of Korea; jdher@kitox.re.kr; 5Department of Animal Science, College of Life Science, Sangji University, Wonju 26339, Republic of Korea; meeahn20@sangji.ac.kr; 6Division of Animal Bioscience and Integrated Biotechnology, Jinju 52725, Republic of Korea; hwkim@gnu.ac.kr

**Keywords:** RNA-sequencing, apoptotic process, Hep3B, liver cancer

## Abstract

*Glehnia littoralis* is a perennial herb found in coastal sand dunes throughout East Asia. This herb has been reported to have hepatoprotective, immunomodulatory, antioxidant, antibacterial, antifungal, anti-inflammatory, and anticancer activities. It may be effective against hepatocellular carcinoma (HCC). However, whether this has been proven through gene-level RNA-seq analysis is still being determined. Therefore, we are attempting to identify target genes for the cell death process by analyzing the transcriptome of Hep3B cells among HCC treated with GLE (*Glehnia littoralis* extract) using RNA-seq. Hep3B was used for the GLE treatment, and the MTT test was performed. Hep3B was then treated with GLE at a set concentration of 300 μg/mL and stored for 24 h, followed by RNA isolation and sequencing. We then used the data to create a plot. As a result of the MTT analysis, cell death was observed when Hep3B cells were treated with GLE, and the IC50 was about 300 μg/mL. As a result of making plots using the RNA-seq data of Hep3B treated with 300 μg/mL GLE, a tendency for the apoptotic process was found. Flow cytometry and annexin V/propidium iodide (PI) staining verified the apoptosis of HEP3B cells treated with GLE. Therefore, an increase or decrease in the DEGs involved in the apoptosis process was confirmed. The top five genes increased were *GADD45B*, *DDIT3*, *GADD45G*, *CHAC1*, and *PPP1R15A*. The bottom five genes decreased were *SGK1*, *CX3CL1*, *ZC3H12A*, *IER3*, and *HNF1A*. In summary, we investigated the RNA-seq dataset of GLE to identify potential targets that may be involved in the apoptotic process in HCC. These goals may aid in the identification and management of HCC.

## 1. Introduction

Globally, hepatocellular carcinoma (HCC) is the most prevalent type of liver cancer and is growing at the fastest rate in terms of cancer-related mortality [1,2]. Chemotherapy, radiation, liver transplantation, surgical resection, and immunotherapy are among the multimodal therapeutic approaches for HCC [3,4]. Low therapeutic profiles in HCC are demonstrated by chemotherapy resistance and post-surgical recurrence [5]. As a result, new approaches to diagnosing and treating HCC are required, as is a deeper understanding of the molecular pathways behind this harmful disease progression [6,7].

*Glehnia littoralis* is a perennial herb found in coastal dunes across East Asia and is a member of the Apiaceae family [8,9]. Radix Glehniae, the root of *G. littoralis*, has long been used in medicine for its analgesic, diaphoretic, and antipyretic effects [10]. It is also used as a tonic and mucolytic for treating gastrointestinal and respiratory problems and is included in the pharmacopoeias of China, Japan, and Korea [11,12]. The dried roots are also helpful as a food ingredient and a therapeutic ingredient for nutritious meals. The roots are frequently used in soups, cereals, medicinal wines, and teas for analysis [13]. There are numerous medical benefits associated with this herb, including hepatoprotective, immunomodulatory, antioxidant, antibacterial, antifungal, anti-inflammatory, and anticancer activities [14,15]. 

Apoptosis, commonly understood to be programmed cell death, is crucial for the establishment and preservation of tissue homeostasis as well as for the chemoprevention of cancer. Multiple unique morphological characteristics indicate apoptosis, including membrane blebbing, cell shrinkage, chromatin condensation, DNA fragmentation, and eventual engulfment by macrophages [16]. *Glehnia littoralis* has been shown to enhance the probability that apoptosis is induced as an anticancer response [14]. However, the mechanism underlying anticancer responses at the RNA-seq analysis level is unknown. Therefore, *Glehnia littoralis* extract (GLE) will be used to treat Hep3B, one of the hepatocellular carcinomas, and RNA-seq will be used to verify the treatment efficacy.

The primary method for transcriptome profiling is becoming RNA-seq. After treatment, transcriptome analysis combined with bioinformatics data mining tools enables the analysis of large genes and targets, as well as the identification of the mechanism of action [17]. RNA-seq is superior to microarray analysis because it can quantify the number of transcripts, including protein-coding genes (mRNA) and long non-coding RNA transcripts (lncRNA), detect novel transcripts, fusion genes, and mutations, define RNA species, and identify rare RNA transcript variants [18,19]. Gene ontology (GO), pathway enrichment analysis, and the functional categorization of gene annotations are examples of downstream RNA-seq analysis [20,21].

This study used RNA-seq transcriptome analysis to determine the relationship between GLE and apoptotic mechanisms in HCC. First, under GLE treatment, we examined the cell survival rate of Hep3B cells. Subsequently, we performed RNA-seq transcriptome analysis to explore the complete HCC transcriptome. Furthermore, we analyzed the connections between the selected genes and those connected to the cell death process.

## 2. Results

### 2.1. Glehnia littoralis Extract (GLE) Induces Cell Death of Hep3B Cells

The MTT assay assessed how GLE affected human Hep3B liver cancer cell mortality. GLE was added to Hep3B cells and incubated for 24 h at concentrations of 0, 10, 20, 50, 75, 100, 200, 300, 400, and 500 μg/mL to assess their impact on cell mortality. As the concentration of GLE erose, cell viability decreased, as seen in Figure 1A. The IC_50_ of GLE was about 300 μg/mL when GLE was treated with Hep3B. This concentration was toxic to Hep3B cells but was also found to be a concentration that could be used to identify the kind of cell death. Therefore, it was decided to employ this concentration in further research.

This study utilized a light microscope to investigate the morphological features of Hep3B cells treated with GLE. Remarkably, compared to control cells, cells treated with 100, 300, and 500 μg/mL GLE for 24 h displayed apoptotic morphological characteristics, including shrinkage and cytoplasmic blebbing (Figure 1B). These findings demonstrated the substantial cytotoxic effect of GLE on Hep3B cells.

### 2.2. Identification of Genes and Differentially Expressed Genes (DEGs)

A total of 1148 DEGs (log2 (Fold Change) > 1.0 and *p*-value < 0.05) were identified in the GLE treatment group, of which 636 upregulated genes and 512 downregulated genes were identified (Table 1). Furthermore, DEGs among all the genes were visible in the scatter plot (Figure 2).

### 2.3. Gene Ontology Term Enrichment Analysis of DEGs in GLE-Treated Hep3B Cells

GO annotation and KEGG pathway enrichment analyses were performed to systematically identify DEGs’ functions and target signaling pathways in GLE-treated cells. GO analysis showed that DEGs were involved in biological processes, cellular components, and molecular functions (Figure 3). In biological processes, response to unfolded protein, PERK-mediated unfolded protein response, and integrated stress response signaling occur most frequently. Among cellular components, nuclear speck, Cul3-RING ubiquitin ligase complex, and inclusion body appear most frequently in that order. Genes related to molecular functions appear in this order: DNA-binding transcription repressor activity, RNA polymerase II-specific, DNA-binding transcription repressor activity, and transcription coregulator activity.

### 2.4. KEGG (Kyoto Encyclopedia of Genes and Genomes) Enrichment Pathway and Protein–Protein Interaction (PPI)

Among the KEGG enrichment plots of data from the KEGG enrichment pathway analysis among all DEGs (Table 2), the apoptotic process (Figure 4) data related to morphology was created as a KEGG enrichment plot. A protein–protein interaction (PPI) network analysis was constructed using data from 64 of the 88 DEGs with increases or decreases in the apoptotic process STRING database, excluding unconnected proteins (Figure 5, Table 3). The results from the PPI network were organized based on the number of network connections (Table 4).

### 2.5. GLE Induces Apoptotic Cell Death in Hep3B Cells

Using flow cytometry and the double staining of propidium iodide (PI) and allophycocyanin (APC)/annexin V, we examined the effect of GLE on the induction of apoptosis in Hep3B cells. At 0, 100, and 300 μg/mL concentrations, GLE treatment increased by 0.31%, 12.93%, and 35.14% in early apoptosis (bottom right quadrant) (Figure 6). Furthermore, at concentrations of 0, 100, and 300 μg/mL, respectively, GLE administration elevated late apoptosis (upper right quadrant) to 0.03%, 7.84%, and 24.52% (Figure 6). The total cell death was considerably increased by 0.34%, 20.77%, and 59.66%, respectively, at these concentrations (Figure 6). These findings demonstrate that GLE caused Hep3B cells to proceed with apoptosis.

### 2.6. Increase or Decrease in Genes Related to the Apoptotic Process

A circular cluster heatmap of protein-coding genes from the Hep3B dataset used in the PPI network is shown in Figure 7. RNAs differentially expressed in Hep3B after GLE treatment are listed in Appendix A.

The five genes that showed a significant increase or decrease among those in the circular cluster heatmap were each obtained in a box plot. The following genes showed a significant increase in order: *GADD45B*, *DDIT3*, *GADD45G*, *CHAC1*, and *PPP1R15A* (Figure 8). The genes that showed the most decrease were *IER3*, *HNF1A*, *ZC3H12A*, *CX3CL1*, and *SGK1* (Figure 9).

## 3. Discussion

Understanding the elements that cause disease provides a roadmap for creating novel therapeutic medications [22]. RNA-sequencing technology, by identifying the targets of numerous diseases, including cancer, is employed for both therapeutic applications and disease diagnosis [23,24,25]. Network analysis is a valuable tool for understanding complex relationships. It often uses pharmacology and bioinformatics to clarify drug–target interaction mechanisms [26,27,28]. This technology means patients may have a broader range of treatment options thanks to targeted medication therapy that targets genes specifically expressed in tumors.

The toxicity of GLE to Hep3B cells was examined. According to our research, GLE dramatically increased Hep3B cell death (Figure 1A). GLE treatment of Hep3B cells was also used to explore the phenotypic traits of these cells using gross morphology microscopy (Figure 1B). Interestingly, the GLE treatment fails to preserve the original cell shape and, in a dose-dependent manner, markedly increases the number of bubble droplets on the plasma membrane. Large plasma membrane bubble droplets accumulated, and Hep3B cells died as a result of 300 μg/mL of GLE, as we discovered. Based on the information, it may be concluded that GLE influences cell death.

The effect of GLE on the genetic changes in Hep3B cells was analyzed using RNA-seq. Gene Ontology changes and 1148 differentially expressed genes (DEGs) were identified between the GLE-non-treated and GLE-treated groups (Figure 2, Table 1). Genes related to DNA binding transcription repressors increased in molecular functions, genes related to cell growth regulation increased in biological processes, and genes related to the nuclear cyclin-dependent protein kinase holoenzyme complex decreased in cellular components (Figure 3, Table 2). GO analysis confirmed that GLE inhibits hepatocellular carcinoma.

Using GO analysis, pathways involved in DEGs were arranged into a table (Table 3). According to the Bg Ratio, apoptosis was the most frequently expressed gene in Table 3, which measures the number of expressed genes. Consequently, the plot (Figure 4) can be used to confirm the level of expression of the genes involved based on the data. These findings suggest a role for GLE in the apoptotic process.

Protein–protein interaction (PPI) network analysis corroborated the association of genes among the 88 DEGs implicated in the apoptotic process, except for 24 genes that showed no correlation (Figure 5). Additionally, it was demonstrated to be connected to several pathways in addition to the apoptotic process (Table 5). This analysis suggests that Hep3B may play a role in apoptosis by linking it to multiple pathways.

The most promising technique for treating malignancies is apoptosis induction, since malignant cells tend to resist apoptosis [29]. One of the mechanisms implicated in tumor cell death is apoptosis, a form of programmed cell death [30]. This study used APC/V and PI double labeling to verify whether GLE causes apoptosis in Hep3B cells. Flow cytometric analysis of GLE-treated Hep3B cells using allophycocyanin (APC)/annexin V and propidium iodide (PI) double staining showed increased fractions of apoptotic cell death in both early and late apoptosis phases (Figure 6). There was a notable increase in the death fraction as well. Therefore, it was visually verified that GLE caused apoptotic cell death in Hep3B cells.

To determine the difference in expression levels of DEGs involved in the apoptotic process before and after GLE treatment, a circular cluster heatmap was drawn using the values in Appendix A (Figure 7). As a result, the five genes whose expression levels increased after GLE treatment were *GADD45B*, *DDIT3*, *GADD45G*, *CHAC1*, and *PPP1R15A*. On the other hand, the five genes whose expression levels decreased after treatment were *SGK1*, *CXECL1*, *ZC3H12A*, *IER3*, and *HNF1A* (Appendix A). Box plots were drawn with five genes, each showing significant increases and decreases, suggesting the existence of candidate gene expression (Figure 8 and Figure 9).

The top five DEGs are *GADD45B*, *DDIT3*, *GADD45G*, *CHAC1*, and *PPP1R15A*, and the bottom five DEGs are *IER3*, *HNF1A*, *ZC3H12A*, *CX3CL1*, and *SGK1*. Among them, *GADD45B* and *GADD45G*, which are among the top five, are thought to be able to draw meaningful conclusions regarding apoptosis. *GADD45B* and *GADD45G* are known as genes that inhibit growth and can cause DNA damage [31]. *GADD45B* activation is mediated through a protein that binds and activates MTK1/MEKK4 kinase, an upstream activator of p38 and JNK MAPK. The function of the GADD45B gene or protein product is involved in the regulation of growth and apoptosis [32]. The protein encoded by the *GADD45G* gene also mediates the activation of the p38/JNK pathway via MTK1/MEKK4 kinase [33]. This shows that, like *GADD45B*, it is related to JNK/MAPK and MTK1/MEKK4, so both genes can be said to be related to apoptosis. Since these two genes are ranked first and second on the list of increased genes, it is possible that they can also be explained molecularly by apoptosis.

According to existing studies, *Glehnia littoralis* inhibits the G1 phase proliferation of human breast cancer MCF-7 cells [34]. It is also effective in inhibiting liver and lung cancers [35], while GLE inhibits the proliferation of human tumor cells [36]. Therefore, based on our study and these known studies, GLE can be considered as a potential cancer therapeutic. Therefore, to investigate the relationship between GLE and hepatocellular carcinoma, transcriptome analysis was performed in Hep3B cells treated with GLE. This investigation identified 64 DEGs related to the apoptosis process. Among them, 10 genes can be used as potential markers for hepatocellular carcinoma if further validation of GLE-induced apoptosis induction is performed. However, cell lines cannot accurately reproduce the in vivo environment, so they are only rough models to help understand the tumor environment, which limits the current study. In this context, additional comprehensive investigations involving multiple cell lines are still needed in the future to ensure diversity and validate the in vivo model.

## 4. Material and Methods

### 4.1. Reagents and Chemicals

The experiment used GLE, an extract from Glehnia littoralis, in 70% methanol [37]. Dulbecco’s modified Eagle’s medium (DMEM), fetal bovine serum (FBS), phosphate-buffered saline (PBS), 0.05% Trypsin-EDTA, and antibiotics penicillin/streptomycin (P/S) were purchased from Gibco (BRL Life Technologies, Grand Island, NY, USA).

### 4.2. Cell Culture and Viability Assay

The American Type Culture Collection (ATCC, Manassas, VA, USA) provided the Hep3B hepatocellular carcinoma cells, which were then grown in full DMEM with 10% FBS and supplemented with 100 U/mL penicillin and 100 µg/mL streptomycin (P/S). The cells were cultured in a humidified environment with 5% CO_2_ at 37 °C.

For 12 h, Hep3B cells were planted in 96-well plates at a density of 1 × 10^4^ cells per well. Next, for 24 h, the cells were exposed to varying doses of GLE (0, 10, 25, 50, 75, 100, 200, 300, 400, and 500 μg/mL). After treating each well with 10 μL of MTT solution (5 mg/mL) and 90 μL of media, the cells were grown at 37 °C for 4 h. It was decided that the insoluble formazan crystals should be dissolved using DMSO. Finally, each sample was run in triplicate, and a microplate reader (BioTek, Winooski, VT, USA) was used to read each well’s absorbance value at 450 nm.

### 4.3. Annexin V/Propidium Iodide Apoptosis Detection

Using an allophycocyanin (APC)/annexin V apoptosis detection kit, apoptotic cells were identified in accordance with the manufacturer’s instructions (BD Biosciences, San Diego, CA, USA). In summary, 3 × 10^7^ cells were plated on each 100 mm plate, and different doses of GLE (0, 100, and 300 μg/mL) were subsequently incubated for 24 h. After being gathered, the cells were rinsed with PBS and added to the binding buffer. Before adding binding buffer, the cells were stained for 15 min at room temperature in the dark using APC/annexin V and propidium iodide (PI). Fluorescence-activated cell sorting (FACSVerseTM flow cytometer; BD Biosciences, Franklin Lakes, NJ, USA) was used to examine the data collected from the flow cytometry examination of the cell suspensions. BD FACSuiteTM software version 1.6 (BD Biosciences, Becton & Dickson, Mountain View, CA, USA) was used to sort and analyze 10,000 events per sample.

### 4.4. RNA Isolation

Total RNA was isolated using Trizol reagent (Invitrogen, Waltham, MA, USA). RNA quality was assessed by the Agilent TapeStation system (Agilent Technologies, Amstelveen, The Netherlands), and RNA quantification was performed using Qubit (Thermo Fisher Scientific Inc., Waltham, MA, USA).

### 4.5. Library Preparation and Sequencing

For control and test RNAs, the library construction was performed using the QuantSeq 3′ mRNA-Seq Library Prep Kit (Lexogen, Inc., Vienna, Austria) according to the manufacturer’s instructions. In brief, each total RNA was prepared, and an oligo-dT primer containing an Illumina-compatible sequence at its 5′ end was hybridized to the RNA, and reverse transcription was performed. After degradation of the RNA template, second strand synthesis was initiated by a random primer containing an Illumina-compatible linker sequence at its 5′ end. The double-stranded library was purified by using magnetic beads to remove all reaction components. The library was amplified to add the complete adapter sequences required for cluster generation. The finished library is purified from PCR components. High-throughput sequencing was performed as single-end 75 sequencing using NextSeq 550 (Illumina, Inc., San Diego, CA, USA).

### 4.6. Data Analysis

QuantSeq 3′ mRNA-Seq reads were aligned using STAR [38]. STAR indices were either generated from the genome assembly sequence or the representative transcript sequences to align with the genome and transcriptome. The alignment file was used to assemble transcripts, estimate their abundances, and detect the differential expression of genes. Differentially expressed genes were determined based on counts from unique and multiple alignments using coverage in HTSeq-count [16]. The RC (read count) data were processed based on TMM+CPM normalization using EdgeR methods. Gene classification was based on searches performed by DAVID (http://davidbioinformatics.nih.gov, access date: 21 January 2024) and Medline databases (http://www.ncbi.nlm.nih.gov, access date: 24 January 2024). Data mining and graphic visualization were performed using ExDEGA (Ebiogen Inc., Seoul, Republic of Korea).

### 4.7. Source Code

The source code for SRPLOT (https://www.bioinformatics.com.cn/srplot, access date: 21 March 2024) [39], STRING (https://string-db.org/cgi/input?sessionId=bceSBoE7p4Rn&input_page_show_search=on, access date: 21 March 2024), GEPIA 2 (http://gepia2.cancer-pku.cn/#index, access date: 18 March 2024), and the ggplot2 package in the R program are used in this study.

## 5. Conclusions

We have identified potential targets that could be involved in the apoptotic process of Hep3B by analyzing the RNA-seq dataset in GLE. The results of this study suggest that more preclinical and clinical trials are necessary to put a treatment plan into action. Still, they may also provide fundamental information for creating novel medications made from natural products and for research into mechanisms.

## Figures and Tables

**Figure 1 ijms-25-09462-f001:**
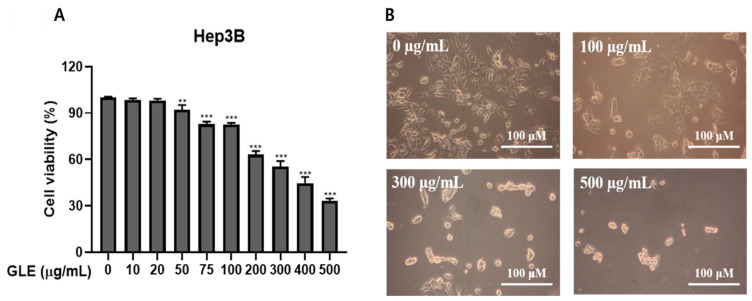
Hep3B cells were subjected to cell death by GLE. (**A**) Cytotoxicity assessment following Hep3B cell treatment with GLE. For 24 h, either 0.1% DMSO or 0, 10, 20, 50, 75, 100, 200, 300, 400, and 500 μg/mL of GLE was applied to Hep3B cells. Using the MTT assay, the cytotoxicity was assessed. The information shows the average percentage of control ± S.D. (n = 3). ** *p* < 0.01, *** *p* < 0.001 vs. control group. (**B**) GLE impacts Hep3B cell morphology (magnification × 100; scale bar 100 μm). For 24 h, the cells were treated with GLE at 0, 100, 300, and 500 μg/mL. Following the application of GLE, a phase-contrast picture of the cells was captured. (n ≤ 3).

**Figure 2 ijms-25-09462-f002:**
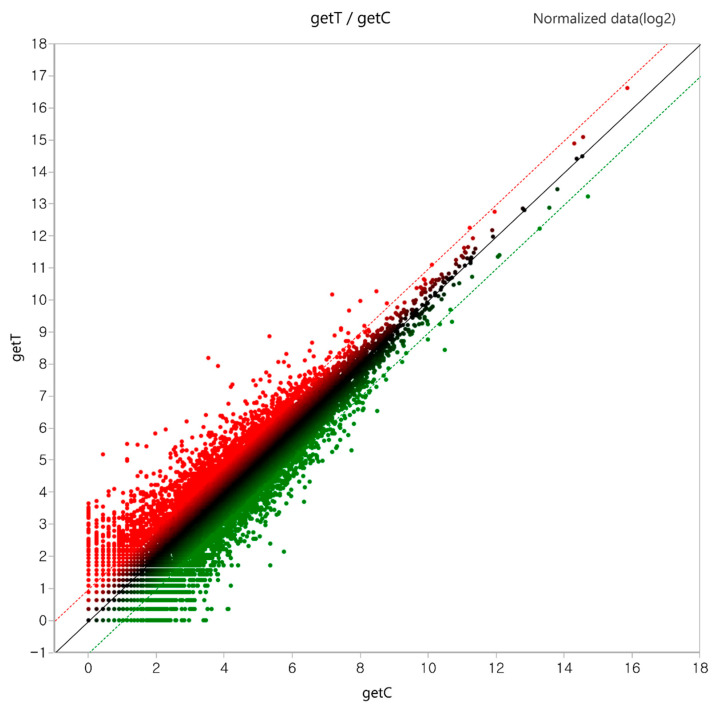
Scatter plot of differentially expressed genes (DEGs). In total, 1148 DEGs represented by getT/getC values were considered in the plot. Normalized data (log2) were used as the values. Red indicates increase and green indicates decrease.

**Figure 3 ijms-25-09462-f003:**
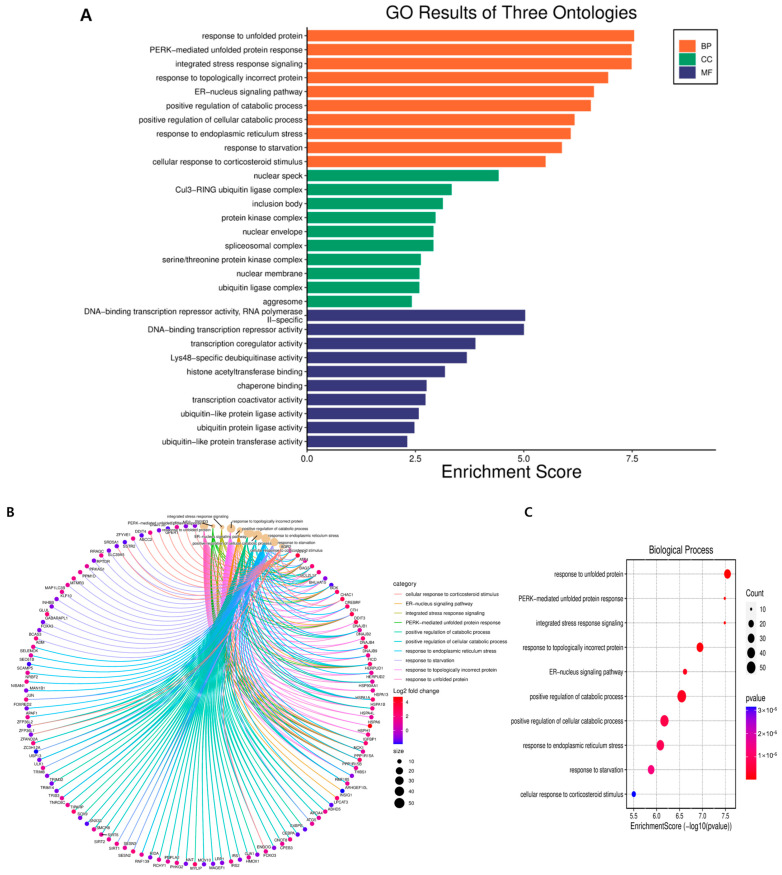
GO terms of DEGs functionally arranged in Hep3B cells treated with GLE. (**A**) Gene Ontology (GO) analysis was used to predict the potential functions of DEGs in biological processes, cellular components, and molecular functions. (**B**) Chord diagram showing what function each gene is connected to in a biological process. (**C**) Bubble plot showing biological processes. (**D**) Chord diagram showing what function each gene is linked to in cellular components. (**E**) Bubble plot showing cellular components. (**F**) Chord diagram showing which function each gene is linked to in molecular function. (**G**) Bubble plot showing molecular function. Genes that showed a >2-fold change in expression and a *p*-value of less than 0.05 were considered differentially expressed.

**Figure 4 ijms-25-09462-f004:**
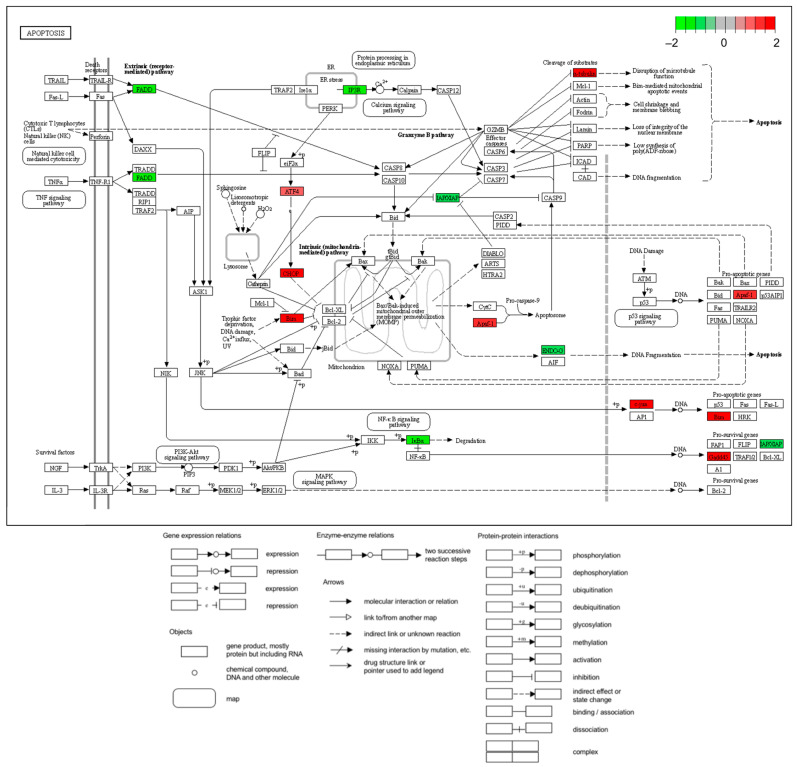
KEGG enrichment pathway analysis. The DE gene list genes are superimposed on the apoptosis pathway. Green: Hep3B genes that were downregulated experimentally. Red: Hep3B genes that are experimentally elevated.

**Figure 5 ijms-25-09462-f005:**
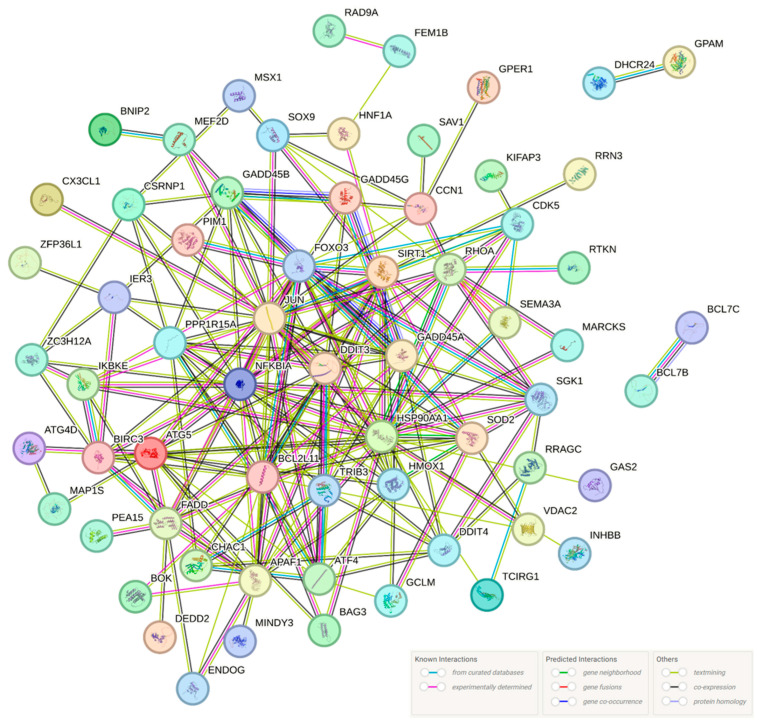
Protein-protein interaction (PPI) analysis of apoptotic process DEGs was analyzed using STRING software version 12.0. The round nodes indicate individual genes, whereas the network nodes represent genes (displaying interactions). The color of each circle represents each protein. The color of the lines indicates the sort of interaction evidence.

**Figure 6 ijms-25-09462-f006:**
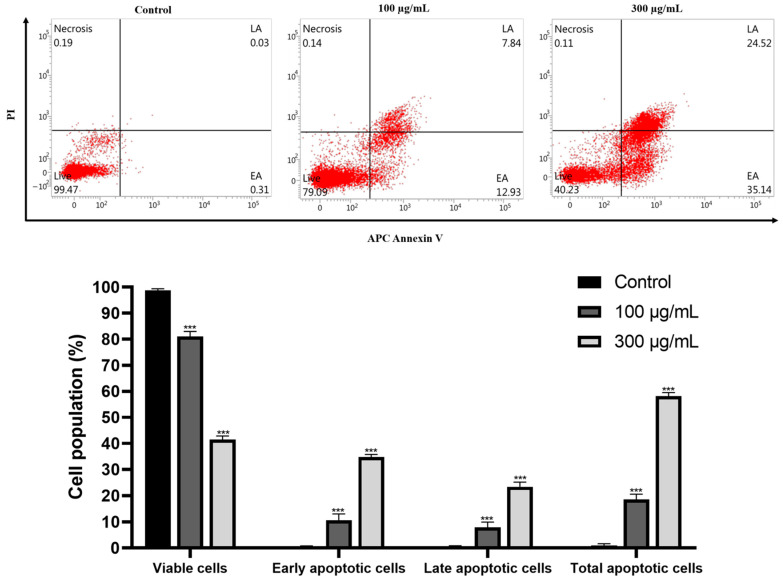
Hep3B cells are subjected to apoptosis by GLE. The cells were subjected to GLE treatment for 24 h at the stated concentrations (0, 100, and 300 μg/mL) to measure the degree of apoptosis induced by GLE. Using propidium iodide (PI) and allophycocyanin (APC)/annexin V double staining, flow cytometry was used for analysis. Each red dot represents a cell. Three separate experiments generated data expressed as the mean ± standard error of the mean (SEM) in relation to the control group. *** *p* < 0.001.

**Figure 7 ijms-25-09462-f007:**
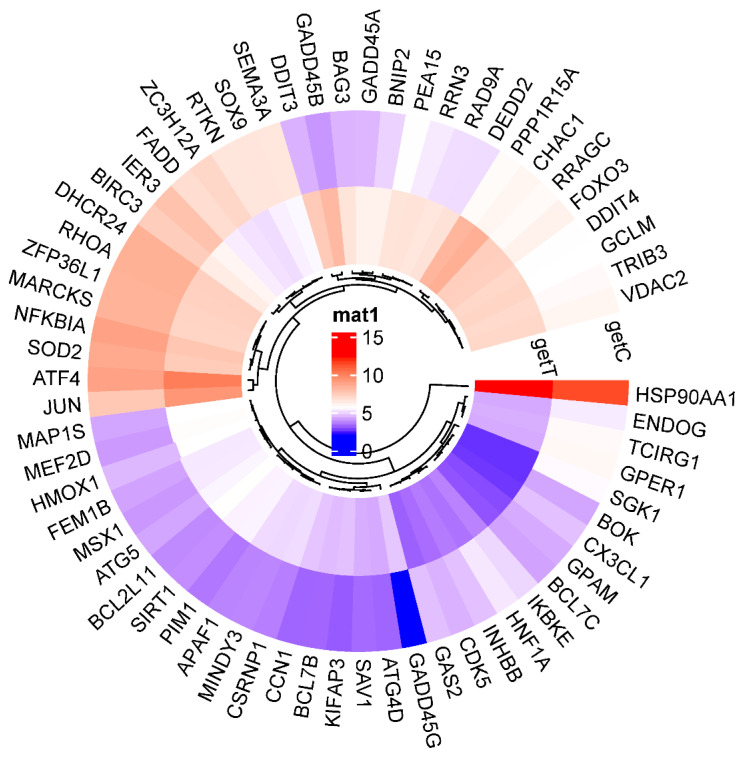
A circular cluster heatmap shows the rise and fall in apoptotic-process-related genes in Hep3B following GLE treatment. Gene expression levels are indicated by red for high expression and blue for low expression.

**Figure 8 ijms-25-09462-f008:**
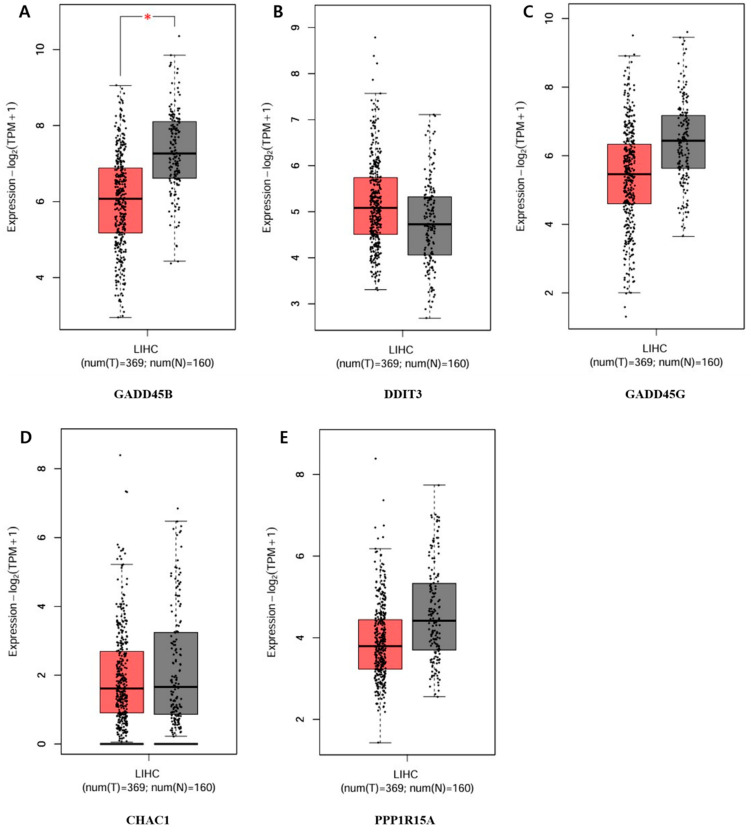
Top-5 genes increasing in the apoptotic process of the PPI network. Validation of the mRNA expression levels of (**A**) *GADD45B*, (**B**) *DDIT3*, (**C**) *GADD45G*, (**D**) *CHAC1*, (**E**) *PPP1R15A* in LIHC tissues and normal liver tissues using GEPIA. Red: tumor color, Black: normal color. * *p* < 0.01 was considered statistically significant. LIHC: Liver hepatocellular carcinoma.

**Figure 9 ijms-25-09462-f009:**
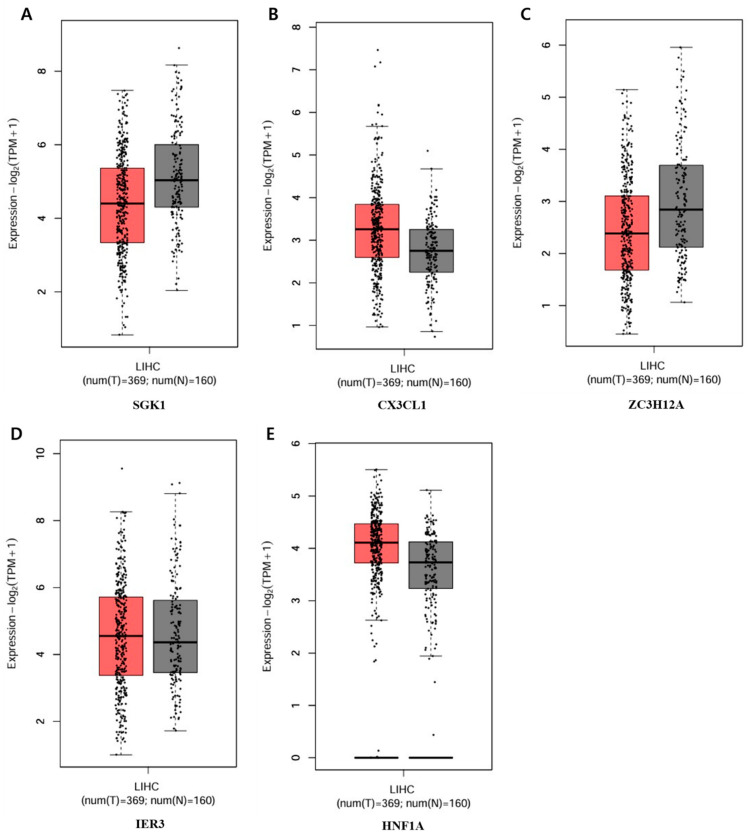
Bottom-5 genes decreasing in the apoptotic process of the PPI network. Validation of the mRNA expression levels of (**A**) *SGK1*, (**B**) *CX3CL1*, (**C**) *ZC3H12A*, (**D**) *IER3*, (**E**) *HNF1A* in LIHC tissues and normal liver tissues using GEPIA. Red: tumor color, Black: normal color. LIHC: Liver hepatocellular carcinoma.

**Table 1 ijms-25-09462-t001:** Sequencing statistical data. getC is the control group in Hep3B that was not treated with GLE. getT is the test group in Hep3B that was treated with GLE.

Name	Type	Reads	Bases	Bases (Gb)	GC	N	Q30
getC	Raw	22,236,783 (100%)	1,689,995,508 (100%)	1.69	722,506,852 (42.75%)	122,257 (0.0%)	1,546,777,236 (91.53%)
	Clean	21,935,961 (98.65%)	1,563,188,663 (92.50%)	1.56	685,071,168 (43.83%)	17,858 (0.0%)	1,471,970,080 (94.16%)
getT	Raw	16,504,954 (100%)	1,254,376,504 (100%)	1.25	525,542,567 (41.90%)	91,010 (0.0%)	1,138,775,855 (90.78%)
	Clean	16,279,626 (98.63%)	1,148,658,184 (91.57%)	1.15	496,909,263 (43.26%)	13,046 (0.0%)	1,075,923,116 (93.67%)

Reads: Number of reads (Reads/Raw reads × 100); Bases: Number of bases (Bases/Raw bases × 100); Bases (Gb): Number of bases (in Giga base unit); GC: Number of G and C bases (GC/Bases × 100); N: Number of N bases (N/Bases × 100); Q30: Number of over Q30 bases (Q30: 99.9% Base Call Accuracy) (Q30/Bases × 100); Q20: Number of over Q20 bases (Q20: 99% Base Call Accuracy) (Q20/Bases × 100).

**Table 2 ijms-25-09462-t002:** RNA-seq functional enrichments.

**Biological Process**				
**ID**	**Description**	**Gene** **Ratio**	**BgRatio**	***p* Value**
GO:0042594	response to starvation	29/994	206/18,866	1.31859 × 10^−6^
GO:0001558	regulation of cell growth	38/994	420/18,866	0.000847655
GO:0006984	ER-nucleus signaling pathway	14/994	51/18,866	2.40565 × 10^−7^
**Molecular Function**				
**ID**	**Description**	**Gene** **Ratio**	**BgRatio**	***p* Value**
GO:0001217	DNA-binding transcription repressor activity	39/1015	336/18,352	9.92372 × 10^−6^
GO:0003712	transcription coregulator activity	48/1015	498/18,352	0.000130717
GO:0004402	histone acetyltransferase activity	8/1015	55/18,352	0.010272488
**Cellular Component**				
**ID**	**Description**	**Gene** **Ratio**	**BgRatio**	***p* Value**
GO:0016607	nuclear speck	41/1029	401/19,559	3.81569 × 10^−5^
GO:0005635	nuclear envelope	41/1029	473/19,559	0.001216517
GO:0019908	nuclear cyclin-dependent protein kinase holoenzyme complex	3/1029	12/19,559	0.022350144

**Table 3 ijms-25-09462-t003:** Gene Ontology term enrichment analysis of genes differently expressed in response to GLE. The top six Gene Ontology (GO) categories that were found to be enriched in the collection of genes that were differentially expressed (DE) in our transcriptome study are displayed in this table. The table includes Gene Ratio (number of DE genes belonging to the GO category over the total number of DE genes), Bg Ratio (number of genes included expressed in Hep3B belonging to the GO category over the total number of genes expressed in Hep3B), pvalue (raw *p*-value), p.adjust (Bonferroni adjusted *p*-values), qvalue (false discovery rate adjusted pvalues), ID (identity of the DE genes included in the GO category), and Count (number of genes in the GO category that are DE).

ID	Description	Gene Ratio	Bg Ratio	*p* Value	*p*.Adjust	q Value	GeneID	Count
hsa04722	Neurotrophin signaling pathway	14/480	119/8223	0.00923833	0.22409163	0.21320458	*ATF4/FOXO3/FRS2/IRAK2/IRS1/JUN/MAP3K3/* *MAPK12/NFKBIA/NFKBIB/NFKBIE/RHOA/RPS6KA5/SOS2*	14
hsa04152	AMPK signaling pathway	12/480	121/8223	0.04947161	0.50476492	0.48024193	*ADIPOR2/FOXO3/HNF4A/IRS1/IRS2/PRKAG1/* *RHEB/RPTOR/SIRT1/STRADB/TBC1D1/ULK1*	12
hsa04110	Cell cycle	13/480	127/8223	0.03374595	0.48461782	0.46107364	*CCNE1/CDC6/CDK7/CDKN1C/CDKN2D/DBF4B/GADD45A/* *GADD45B/GADD45G/ORC1/ORC3/ORC6/TTK*	13
hsa04068	FoxO signaling pathway	16/480	131/8223	0.00387417	0.13927125	0.13250504	*BCL2L11/CDKN2D/FOXO3/GABARAPL1/GADD45A/GADD45B/* *GADD45G/IRS1/IRS2/MAPK12/PRKAG1/SGK1/SGK2/SIRT1/SOD2/SOS2*	16
hsa04210	Apoptosis	14/480	136/8223	0.02708404	0.47755381	0.45435282	*APAF1/ATF4/BCL2L11/BIRC3/DDIT3/ENDOG/FADD/GADD45A/* *GADD45B/GADD45G/ITPR2/JUN/NFKBIA/TUBA4A*	14

**Table 4 ijms-25-09462-t004:** Network stats of the PPI network.

Number of nodes: 64	Avg. local clustering coefficient: 0.678
Number of edges: 197	Expected number of edges: 47
Average node degree: 6.16	PPI enrichment *p*-value: <1.0 × 10^−16^

**Table 5 ijms-25-09462-t005:** Functional enrichments in the PPI network.

**Biological Process**				
**GO-Term**	**Description**	**Count in Network**	**Strength**	**False Discovery Rate**
GO:0006915	Apoptotic process	64 of 1041	1.28	1.87 × 10^−77^
GO:0050794	Regulation of cellular process	61 of 11,025	0.23	5.23 × 10^−10^
GO:0050896	Response to stimulus	54 of 7835	0.33	9.81 × 10^−11^
GO:0051716	Cellular response to stimulus	52 of 6357	0.4	6.42 × 10^−13^
GO:0048518	Positive regulation of the biological process	51 of 6207	0.4	1.61 × 10^−12^
**Molecular Function**				
**GO-Term**	**Description**	**Count in Network**	**Strength**	**False Discovery Rate**
GO:0005515	Protein binding	47 of 7242	0.3	7.04 × 10^−6^
GO:0019899	Enzyme binding	28 of 2084	0.62	6.33 × 10^−8^
GO:0042802	Identical protein binding	20 of 2144	0.46	0.0093
GO:0098772	Molecular function regulator activity	19 of 1960	0.47	0.0093
GO:0044877	Protein-containing complex binding	15 of 1261	0.56	0.0093
**KEGG Pathways**				
**Pathway**	**Description**	**Count in Network**	**Strength**	**False Discovery Rate**
hsa05200	Pathways in cancer	13 of 515	0.89	1.29 × 10^−6^
hsa04210	Apoptosis	12 of 131	1.45	1.03 × 10^−11^
hsa04621	NOD-like receptor signaling pathway	9 of 173	1.2	1.17 × 10^−6^
hsa05169	Epstein–Barr virus infection	9 of 192	1.16	1.29 × 10^−6^
hsa04068	FoxO signaling pathway	8 of 126	1.29	1.29 × 10^−6^
**Reactome Pathways**				
**Pathway**	**Description**	**Count in Network**	**Strength**	**False Discovery Rate**
HSA-162582	Signal transduction	25 of 2540	0.48	4.14 × 10^−5^
HSA-74160	Gene expression (Transcription)	17 of 1476	0.55	0.00080
HSA-168256	Immune system	17 of 1979	0.42	0.0136
HSA-2262752	Cellular responses to stress	15 of 747	0.79	1.69 × 10^−5^
HSA-212436	Generic transcription pathway	15 of 1215	0.58	0.0014
**Wiki Pathways**				
**Pathway**	**Description**	**Count in Network**	**Strength**	**False Discovery Rate**
WP1772	Apoptosis modulation and signaling	10 of 89	1.54	5.91 × 10^−10^
WP3888	VEGFA-VEGFR2 signaling	10 of 428	0.86	0.00012
WP254	Apoptosis	7 of 83	1.41	6.30 × 10^−6^
WP4754	IL-18 signaling pathway	7 of 271	0.9	0.0015
WP2882	Nuclear receptors meta-pathway	7 of 312	0.84	0.0024

## Data Availability

The data used to support the findings of this study are available upon request from the corresponding author.

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
