# Peer review of "Correlation with Apoptosis Process through RNA-Seq Data Analysis of Hep3B Hepatocellular Carcinoma Cells Treated with Glehnia littoralis Extract (GLE)"

_ijms, 2024, doi:10.3390/ijms25179462_

Round 1

Reviewer 1 Report

Comments and Suggestions for Authors

A study by Park et al. investigates the activity of Glehnia littoralis extract in hepatocellular carcinoma cells. In general, this study requires further experimental validation to substantiate the conclusions.

1. The part of title "... reveals Target Genes and Apoptotic Process" is unclear. First of all, RNA-seq revealed changes in gene expression which does not mean that these genes are targeted by the active compound directly. Secondly, "apoptotic process" is also unclear. Did the Author mean "genes associated with apoptotic process"?

2. Fig. 1B lacks scale bars.

3. The major concern relates to the lack of experimental validation of candidate genes. Assessing the fraction of apoptotic cells in response to GLE treatment although demonstrates that apoptosis is actually induced, it does not validate candidate gene roles in this process. In this respect, the manuscript is largely descriptive.

4. English should be revised thoroughly.

Comments on the Quality of English Language

requires revision

Author Response

Reviewer 1

A study by Park et al. investigates the activity of Glehnia littoralis extract in hepatocellular carcinoma cells. In general, this study requires further experimental validation to substantiate the conclusions.

  1. The part of title "... reveals Target Genes and Apoptotic Process" is unclear. First of all, RNA-seq revealed changes in gene expression which does not mean that these genes are targeted by the active compound directly. Secondly, "apoptotic process" is also unclear. Did the Author mean "genes associated with apoptotic process"?

Answer: Thank you for your advice. I have revised the title based on your comments. To your second question, I would say that it means "genes associated with the apoptotic process."

  1. Fig. 1B lacks scale bars.

Answer: Thank you for your advice. I added scale bars in Fig. 1B

  1. The major concern relates to the lack of experimental validation of candidate genes. Assessing the fraction of apoptotic cells in response to GLE treatment although demonstrates that apoptosis is actually induced, it does not validate candidate gene roles in this process. In this respect, the manuscript is largely descriptive.

Answer: Thank you for your advice. According to NCBI and KEGG, among the candidate genes I introduced, GADD45B and GADD45G are genes that induce growth arrest and DNA damage. There is also a paper that says that GADD45 genes play a related role (1). In the current situation, including today, the day of writing this revision, there are only three days left until graduation, so I ask for your understanding that I can no longer proceed with the experiment. I plan to advance to a doctoral program after graduating from the master's program. At that time, I plan to continue the research on the things you indicated.

(1) Humayun, Arslon, and Albert J. Fornace Jr. "GADD45 in stress signaling, cell cycle control, and apoptosis." Gadd45 Stress Sensor Genes. Cham: Springer International Publishing, 2022. 1-22.

GADD45B (https://www.ncbi.nlm.nih.gov/gene/4616)

GADD45G (https://www.ncbi.nlm.nih.gov/gene/10912#gene-expression)

  1. English should be revised thoroughly.

Answer: Thank you for your advice. I rechecked the English and had my lab mates read it to me.

Reviewer 2 Report

Comments and Suggestions for Authors

In the current study, Min Yeong Park et al. investigated the potential therapeutic effects of Glehnia littoralis Extract (GLE) for hepatocellular carcinoma and underlying molecular targets of GLE. To this aim, they studied the human hepatocellular carcinoma cell line, Hep3B treated by GLE and performed comprehensive transcript analysis. The primary weakness of the current manuscript is the study design of HepG3 treatment. The authors used 300 ug/mL GLE based on the IC50 value obtained from the MTT assay. However, it is unclear whether this dose has selective cell toxicity for HCC as described below. Further, several points listed below need to be improved.

Major comments:

·      The authors describe that 300 ug/mL of GLE, which demonstrated IC 50 for HepG3 cells (Figure 1A), was selected to study potential molecular targets of GLE causing cell death. First, an MTT assay employing only Hep3B is not a sufficient assessment for a potential clinical aspect of GLE for treating HCC. If normal hepatocyte cell lines also show IC50 with 300 ug/mL GLE, it is possible that GLE causes cell death even in non-tumor cells, which is an unwanted effect in the clinical setting. Further, it is also possible that 300 ug/mL GLE induces cell death via non-specific molecular targets regardless of cell type. To approach these points, the authors need to perform an MTT assay using the normal hepatocyte cell line treated with the same dose range of GLE. Alternatively, if there are previous publications using GLE for a non-tumor (hepatocyte) cell line, they need to introduce the data to discuss the sensitivity of GLE for HCC.

·      Does GLE also possibly decrease the viability of tumor cells via the arrest of proliferation if different doses were used? Please discuss this point based on previous publications.

·      Phytochemicals contained in natural compounds tend to process multiple molecular targets, leading to therapeutic effects. Given this point, the RNA-seq performed in the current study is valuable in the field. However, the current manuscript does not sufficiently discuss specific molecular targets, possibly multiple targets, those of which are likely responsible for apoptosis caused by 300 ug/mL of GLE. Listing the top five upregulated/downregulated DEGs found in HepG3 treated with GLE in the discussion seems to be descriptive. Please discuss how these DEGs could be involved in molecular events, causing apoptosis.

Minor comments:

·      Please discuss which phytochemicals contained in GLE extract possibly induce cell apoptosis in HCC.

·      Figure 3 A and Figures 3 C, E, and G seem to be repetitive.

·      The letters in Figures 3 B, D, and F are too small and invisible.

·      Please spell out getC and getT.

·      Please consider deposit RNA-seq data.

Comments on the Quality of English Language

Moderate editing of English language required.

Author Response

Dear Reviewer 2

There are figures in my answers, therefore, I would like to respectfully request you to review the attached file. Thank you for your advice and asking.

Round 2

Reviewer 1 Report

Comments and Suggestions for Authors

The comments have been sufficiently addressed.

Comments on the Quality of English Language

minor revision still required

Author Response

Thank you for your advice. I checked English again with lab coworkers and grammar tools.

Reviewer 2 Report

Comments and Suggestions for Authors

Although the authors provided the additional data and discussion points in their responses, the manuscript has not been sufficiently revised. The following points need to be improved.

(1) It seems the authors did not understand the point of the Major comment (1) well. The additional data of the MTT assay included in the response comment seem to indicate that Hep3G is more resistant than none-tumor cell lines (i.e., RAW264.7 cells), while Hep3G and HepG2 show a pretty similar response to GLE exposure. Thus, these data point toward that a higher dose range of GLE is required to induce cell apoptosis in hepatoma cells than that in non-tumor cells, which is not assumed to be a desired therapeutic effect of GLE in the setting of cancer treatment. The authors need to reconsider which original or published data can support the potential clinical application of GLE for hepatoma cellular carcinoma.

(2) The authors need to incorporate the responses for the Major comments (2), (3), and the Minor comment (1) in the revised manuscript. These additional insights and reference publications could help readers understand. For the Minor comment (1), please use previous publications as a reference to discuss potential phytochemicals contained in GLE that could induce cell apoptosis in hepatocarcinoma cells.

(3) The edited versions of Figures 3 B, D, and F are likely to be distorted. Please re-edit them.

Comments on the Quality of English Language

Moderate editing of English language required.

Author Response

Reviewer 2 round 2

Although the authors provided the additional data and discussion points in their responses, the manuscript has not been sufficiently revised. The following points need to be improved.

(1) It seems the authors did not understand the point of the Major comment (1) well. The additional data of the MTT assay included in the response comment seem to indicate that Hep3G is more resistant than none-tumor cell lines (i.e., RAW264.7 cells), while Hep3G and HepG2 show a pretty similar response to GLE exposure. Thus, these data point toward that a higher dose range of GLE is required to induce cell apoptosis in hepatoma cells than that in non-tumor cells, which is not assumed to be a desired therapeutic effect of GLE in the setting of cancer treatment. The authors need to reconsider which original or published data can support the potential clinical application of GLE for hepatoma cellular carcinoma.

Answer: Thank you for your advice. As you mentioned, I have found and added to the discussion a previously published paper on the relationship between Glehnia littoralis and hepatoma cells.

(2) The authors need to incorporate the responses for the Major comments (2), (3), and the Minor comment (1) in the revised manuscript. These additional insights and reference publications could help readers understand. For the Minor comment (1), please use previous publications as a reference to discuss potential phytochemicals contained in GLE that could induce cell apoptosis in hepatocarcinoma cells.

Answer: Thank you for your advice. As you said, I added major 2, 3 contents to the discussion. However, I decided that adding major 1 did not fit the flow of the content in my paper. The focus is on how the integrated extract of GLE acts on Hep3B, not on how each of them acts. As I mentioned in my pharmaceuticals-3167686 paper, I expected a synergistic effect by gathering the existence of each component and its individual potential at once, but did not focus on the individual effects.

(3) The edited versions of Figures 3 B, D, and F are likely to be distorted. Please re-edit them.

Answer: Thank you for your advice. I edited Figure 3 B, D, and F size.

PS. I checked English again with lab coworkers and grammar tools. And I would like to express my deepest gratitude to you for your sincere review. Thanks to your thorough review, I was able to see what was lacking in my thesis and what I should do more in the future.
